# Spatial Dynamics of Demersal Fish Assemblages off the Korean Coast in the East Sea

**DOI:** 10.3390/ani14111612

**Published:** 2024-05-29

**Authors:** Joo Myun Park, Chung Il Lee, Jong Won Park, Hae Kun Jung, In Seong Han

**Affiliations:** 1Dokdo Research Center, East Sea Research Institute, Korea Institute of Ocean Science & Technology, Uljin 36315, Republic of Korea; 2Department of Marine Bioscience, Gangneung-Wonju National University, Gangneung 25457, Republic of Korea; leeci@gwnu.ac.kr (C.I.L.); po7639@gwnu.ac.kr (J.W.P.); 3Fisheries Resources and Environment Research Division, East Sea Fisheries Research Institute, National Institute of Fisheries Science, Gangneung 25435, Republic of Korea; hkjung85@korea.kr; 4Oceanic Climate & Ecology Research Division, National Institute of Fisheries Science, Busan 46083, Republic of Korea; hisjamstec@korea.kr

**Keywords:** demersal fish assemblage, oceanic current, depth layer, East Sea, water temperature

## Abstract

**Simple Summary:**

This study investigated how different fish species are distributed in the sea based on where they are relative to the oceanic current region and how deep the water is. Researchers collected fish samples over four seasons from May 2018 to March 2019 along the coasts of the East Sea in Korea. They used special commercial gill nets to catch the fish at different depths, ranging from shallow to deeper waters. Overall, they found 73 different fish species and collected over 6000 specimens. Depending on the oceanic current region, they found different fish species in each study site. For example, in one area called Ayajin, they mostly found fish like blackfin flounder (*Glyptocephalus stelleri*), Pacific cod (*Gadus macrocephalus*) and mailclad sculpin (*Icelus cataphractus*). In another area called Hupo, different fish species were more common, such as blackfin flounder, Sohachi flounder (*Cleisthenes pinetorum*), and flathead flounder (*Hippoglossoides dubius*). The researchers also noticed that the types of fish varied depending on how deep the water was. This study helps us understand how fish populations change in different parts of the sea based on the oceanic current region and at different depths.

**Abstract:**

This study assessed variations in demersal fish assemblages with respect to the study site and water depth. Seasonal samplings from May 2018 to March 2019 were conducted along the northern (Ayajin, Goseong) and southern (Hupo, Uljin) sites of the East Sea off the Korean coast, using commercial gill nets. Samples were collected at depths of ~50, ~80, ~150 m across the study sites, with concurrent monitoring of water column structures. A total of 73 species and 6250 specimens were collected. Distinctive fish species compositions were observed according to the study site and depth. Although *Glyptocephalus stelleri* was the most abundant fish species in both Ayajin and Hupo, *Gadus macrocephalus*, *Icelus cataphractus*, and *Alcichthys elongatus* were most predominant in Ayajin, whereas *Cleisthenes pinetorum*, *Hippoglossoides dubius*, and *Gymnocanthus herzensteini* were more prevalent in Hupo. In terms of depth layer, in Ayajin, *G. stelleri* dominated in both intermediate and deeper layers, with *Hemilepidotus gilberti*, *A. elongatus*, *Enophrys diceraus* common in shallower depths. Conversely, in Hupo, *G. stelleri*, *C. pinetorum*, and *A. nadeshnyi* dominated across all depth layers, whereas *Dasycottus setiger* and *G. herzensteini* dominated in deeper and shallower depths, respectively. Significant influences of the study site and water depth on fish assemblage structures were observed due to variations in water temperature at the seasonal thermocline boundary.

## 1. Introduction

Understanding the spatial and temporal variations in fish assemblages is important for the effective conservation and management of fisheries resources [1,2]. This knowledge provides a foundation for adopting an ecosystem-based approach to evaluate and manage fisheries resources and biodiversity conservation using spatial management units [3]. In the coastal habitats of temperate seas, the spatial distribution of fish assemblages is largely influenced by oceanic current and/or depth in relation to water temperature and geographical features [4,5,6]. These variations are particularly influenced by factors such as the sediment composition of habitats, physical water column structures, and prey distribution [7,8]. Temporal variations in fish assemblages are associated with changes in environmental variables, such as water temperature, salinity, variability in the use of the local habitat, and migration of fish species [9,10,11].

Marine environments, such as water temperature and salinity in the East Sea off the Korean coast, exhibit high variability depending on the latitudinal path of oceanic currents [12]. Notably, latitudinal patterns of water temperature are evident in the surface layers influenced by two major oceanic currents: the East Korea Warm Current (EKWC) flowing into the East Sea via the Korean Strait, and the North Korea Cold Current (NKCC) branched from the Liman Cold Current (LCC) flowing southward along the eastern coast of the Korean Peninsula from the north [12,13,14]. These interactions divide marine ecosystems in the southwestern part of the East Sea into cooler northern and warmer southern sections [15]. The fish fauna in these regions also demonstrate distinct latitudinal patterns [16]. For example, demersal fish assemblages at the northern limit of South Korea are dominated by boreal fish species, including Pacific cod (*Gadus macrocephalus*) and cold-water flounders, whereas abundant demersal fish from the southeastern coast consist mainly of temperate and some subtropical fish species [17,18]. Consequently, fish fauna in the northern part of the eastern Korean coast exhibit distinctly different species compositions compared with the southern and southeastern fish assemblages influenced by the Tsushima Warm Current (TWC) [19]. Marine communities in the East Sea can be vertically divided into shallower and deeper groups owing to the different influences of the East Korea Warm Current in the surface layer and the North Korea Cold Current in the deeper layer [20].

Over the past six decades, the East Sea has undergone significant environmental changes [21], particularly driven by recent climate warming, resulting in a substantial increase in surface water temperatures [22]. These temperature shifts have triggered alterations in the community structure [23,24], leading to gradual poleward shifts in the distributional ranges of marine organisms, including fishes and invertebrates [25,26], and the emergence of tropical and/or subtropical marine species in the Korean Peninsula [27,28]. However, the lack of comparative studies across different time periods limits our ability to understand historical data and predict future variations.

In this study, we compared demersal fish assemblages in coastal habitats across the study site, depth, and season in the East Sea off the Korean coast. Our specific objectives were to (1) compare species richness, abundance, and diversity between study sites across different depth layers; (2) explore differences in assemblage structures based on study sites, seasons, and depth layers; (3) examine the relationship between changes in demersal fish assemblage structures and water temperature. The findings from this study enhance our understanding of the interaction between water temperature and the spatial structure of demersal fish assemblages in the Korean East Sea, thereby promoting improved conservation and management of fishery resources.

## 2. Materials and Methods

### 2.1. Study Area

Investigations were conducted at two latitudinal locations on the mid-eastern coast of Korea (Figure 1), situated within the southwestern East Sea (also known as the Sea of Japan). This semi-enclosed marginal sea is surrounded by the Korean Peninsula, Russia, and the Japanese Islands, with its oceanographic structures primarily influenced by the Tsushima Warm Current and the Liman Cold Current. Variations in water mass between the warm surface of Tsushima warm water and the homogeneous cold deep water play a significant role in shaping marine ecosystems [29]. Notably, the marine environment of the study area is influenced by two sub-currents, the East Korea Warm Current and the North Korea Cold Current, which branch from the Tsushima Warm Current and the Liman Cold Current, respectively. The two study sites (southern Hupo and northern Ayajin) were selected to identify how the distribution of fish assemblages varied depending on the path of the sub-currents and water depth in the East Sea. The Hupo area is located on the path of the East Korea Warm Current, whereas the Ayajin is located on the boundary between the East Korea Warm Current and the North Korean Korea Cold Current (Figure 1). Sea surface water temperatures in the study area typically ranged from 7.2 to 25.4 °C at the northern site (Ayajin) and from 10.5 to 24.4 °C at the southern site (Hupo) [30].

### 2.2. Sampling

Field sampling was conducted over a year, seasonally from May 2018 to March 2019, using bottom gill nets (75 m long, 2 m high, and 90 mm mesh) at two different locations (Ayajin, northern site, and Hupo, southern site). Four seasons were considered: spring (May–June), summer (August), autumn (October–December), and winter (February–March). Fish samples were collected by installing bottom gill nets for 24 h at three different depth layers (shallow, ~50 m; intermediate, ~80 m; and deeper, ~150 m at Ayajin and ~120 m at Hupo). Immediately after capture, individual fish were preserved on ice and transferred to the laboratory. In the laboratory, the specimens were identified to the species level and weighed to the nearest milligram. All scientific names were checked with reference to FishBase [31]. To determine the vertical water temperature profiles, CTD (conductivity, temperature, and depth) observations were conducted using onboard SBE19 plus during spring, summer, autumn, and winter at each sampling occasion. Because the distance between three sampling locations covering three depth layers is considerably close, CTD observations were conducted only at the deepest location (the outermost station).

### 2.3. Data Analyses

As our sampling involved deploying a single set of fishing gear for 24 h at each site, depth, and season, we did not replicate our sampling efforts. Consequently, our dataset is unsuitable for three-way (site × depth × season) community-level data analyses due to the absence of replication at the lowest factor level. Before conducting data analyses, we assessed for any significant differences in fish assemblage structure between two sites, among three depth layers, and among four seasons using one-way analysis of similarities (ANOSIM). Significant effects were observed at the “site” (global-R = 0.271, *p* = 0.001) and “depth” (global-R = 0.246, *p* = 0.001) factors but not at the “season” (global-R = 0.002, *p* = 0.432). Therefore, we focused our analyses on the two factors: “site” and “depth”.

To estimate community-level diversity, we used the Shannon–Wiener index (H’) [31]. Spatial differences in species richness (number of species), abundance (number of individuals), and diversity were assessed using two-way analysis of variance (ANOVAs) with the Shannon–Weiner index as the response variable. Post hoc ANOVA comparisons were conducted using Tukey’s honestly significant difference test. Prior to the ANOVA test, the abundances of each species were log-transformed (log[abundance + 1]).

Inferential and descriptive analyses were performed to examine abundance trends with the study site and water depth. Permutation multivariate analyses of variance (PERMANOVA) based on Bray–Curtis similarity matrices were conducted on a log (abundance + 1) basis [32]. PERMANOVA is a non-parametric method that uses permutation procedures to test hypotheses and provides a robust and flexible method for assessing differences between groups while accounting for the multivariate nature of the data and the complex experimental designs often encountered in ecology [33,34]. The analysis factors were site (Ayajin and Hupo) and depth (shallow, intermediate, and deeper). The similarity matrices were subjected to a two-way PERMANOVA to test for factor effects. Metric multidimensional scaling (mMDS) ordination was used to visualize the factor effects and to identify how much multivariate dispersions were overlapped among factor groups. Canonical analysis of principal coordinates (CAPs) was used to assess statistical significance among the factor levels and to identify possible contributions of individual species to differences among groups [35,36]. The relative contributions of species to observed differences were assessed using correlation coefficients, and species meeting specific criteria were plotted. Individual species with correlations higher than 0.4 and a total abundance greater than 1 percent were plotted on CAP axes 1 and 2 for additional visualization of the results.

ANOVA was performed using SYSTAT software (IBM SPSS Statistics v28, Chicago, IL, USA). Multivariate analyses were performed using routines in the PRIMER v7 multivariate statistics package (www.primer-e.com) and the PERMANOVA+ add-on module [35,37]. Statistical significance was set at *p* < 0.05.

## 3. Results

### 3.1. Vertical Trends of Water Temperature in Northern and Southern Sites

Water temperatures exhibited distinct seasonal and depth-related patterns at both study sites (Figure 2). Sea surface water temperatures ranged from 5.7 to 22.5 °C in Ayajin and 14.1 to 26.1 °C in Hupo, with clear seasonal variations. In Ayajin, temperatures in the shallow depth layer (~50 m) sharply declined compared to the surface layer, ranging from 3.5 to 9.6 °C, whereas those in the intermediate and deeper layers remained consistently low regardless of season (below 4 °C). Similarly, in Hupo, temperatures in the shallow depth layer decreased with depth, except during autumn when water temperatures remained constant from the surface to a depth of ~60 m. Water temperatures at intermediate depths were below 4 °C except in autumn (7.8 °C) and dropped to between 1.3 and 2.8 °C at deeper depths. These trends in vertical water temperature resulted in a stratified water mass between the shallow and intermediate or deeper layers in both Ayajin and Hupo.

### 3.2. Fish Species Composition

A total of seventy-three species and 6250 specimens of demersal fish from 26 families were collected from the study area (Appendix A). The predominant families, ranked by species number, were Pleuronectidae (13 species), Cottidae (nine species), Sebastidae (eight species), and Liparidae (five species). Species richness was higher in Ayajin, while the fish abundance was higher at Hupo (Appendix A). The most abundant fish families were Pleuronectidae, Cottidae, and Gadidae in both Ayajin and Hupo, whereas Sebastidae and Clupeidae were more abundant in Ayajin and Hupo, respectively (Table 1). In terms of fish species, *Glyptocephalus stelleri* was the most dominant fish species at both Ayajin and Hupo, followed by *Gadus macrocephalus* and *Icelus cataphractus* in Ayajin, and *Cleisthenes pinetorum* and *Hippoglossoides dubius* in Hupo (Table 1).

Demersal fish assemblages showed distinct family compositions in Ayajin and Hupo across the three depth layers (Figure 3). Pleuronectidae fishes comprised the predominant fish group, accounting for >50% of the total fish abundance at all depth layers in Ayajin and Hupo, except at the shallow depth layer in Ayajin, where Cotiidae fishes were the most abundant, followed by Pleuronectidae, Sebastidae, and Hexagrammidae fishes. Cotiidae fishes were the secondary or tertiary most abundant fish groups at intermediate and deeper layers in Ayajin and shallow and intermediate depth layers in Hupo. Gadidae fishes, predominantly *G. macrocephalus*, were abundant only in the intermediate and deeper depth layers at both sites, whereas Clupeidae fishes were more prevalent in the shallow depth layer in Hupo. Psychrolutidae fishes were absent in shallow depth layers but showed higher abundance at deeper depths in both Ayajin and Hupo.

### 3.3. Variations in Species Richness, Abundance, and Diversity

The mean species richness (number of species), abundance (number of specimens), and diversity varied by site and depth. Two-way ANOVA showed that the species richness of the demersal fish assemblages differed significantly between the two study sites and among the three depth layers, whereas abundance and diversity were only significant according to site and depth, respectively (Table 2). No significant two-way interactions were found between the two factors for any of the dependent variables (Table 2).

Tukey’s post hoc tests indicated that the mean species richness and diversity were higher in the shallow depth layer than in the intermediate and deeper habitats, whereas no significant difference was observed among the depth layers for abundance (Figure 4). In addition, the mean species richness and diversity tended to be higher in Ayajin, whereas the mean abundance was higher in Hupo (Figure 4).

### 3.4. Fish Assemblage Structure

A two-way PERMANOVA revealed that fish assemblages were significantly associated with site and depth, and the two-way interaction was significant (*p* < 0.05). Pairwise comparisons of site and depth showed significant differences in fish assemblage structures at shallow depths between Ayajin and Hupo, but no significant between-site differences were observed in the intermediate or deeper layers (Table 3). Significant differences were also observed between the shallow and intermediate-depth layers at Ayajin and between the shallow and deep layers at both Ayajin and Hupo (Table 3).

Metric MDS ordination of the similarity of the mean fish assemblages showed a clear difference by study site and depth (Figure 5). Samples from different sites showed distinct clustering patterns along the MDS2 axis, whereas the multivariate dispersions slightly overlapped with the depths within each site, except for the shallow depth layer in Ayajin (Figure 5). In addition, samples of depth-dependent fish assemblages within each site were clearly divided into three depth layers along the MDS1 axis in the mMDS ordination.

To further investigate the PERMANOVA results, CAP analysis was performed for significant interactions. The CAP plot for site–depth interaction showed a clear separation among the factor groups (Figure 6). Samples from the shallow layer in Ayajin were strongly separated from other sites along the CAP1 axis, and three fish species (*Alcichthys elongatus*, *Enophrys diceraus*, and *Sebastes taczanowskii*) contributed to this separation. *Clupea pallasii*, *Gymnocanthus herzensteini*, *C. pinetorum*, and *Pseudopleuronectes herzensteini* characterized the fish assemblages in the shallow and intermediate depth layers in Hupo, and *Dasycottus setiger*, *G. macrocephalus*, and *G. stelleri* showed species contributions to the intermediate and deeper layers in Ayajin and deeper layers in Hupo (Figure 6).

## 4. Discussion

Among the 73 species collected, three flounder (*G. stelleri*, *C. pinetorum*, and *H. dubius*), one sculpin (*G. herzensteini*), and one Pacific cod (*G. macrocephalus*) emerged as the most dominant species in both Ayajin and Hupo. In addition, four families (Pleuronectidae, Cottidae, Gadidae, and Clupeidae) constituted abundant fish groups. Notably, the dominant fish families and species in our study differ significantly from those in the southeastern and southern coastal regions of the Korean Peninsula [18,38,39]. These species are crucial fisheries resources harvested through gill nets and bottom trawling in the East Sea of Korea [40,41], yet they are rarely landed or are almost absent in the fisheries catches of the Southern Sea of Korea [19,42]. In particular, blackfin flounder (*G. stelleri*) is distributed throughout the high-latitude Northwest Pacific and constitutes the most abundant fishery resource among flounder species in the East Sea [43]. Pacific cod is widely found throughout the North Pacific region and on the Korean Peninsula. The population of this species in Korean waters is mainly distributed in the East Sea; however, a small population is present in the Yellow Sea (western sea area of Korea) [42]. In addition, although the abundances of walleye pollock (*Gadus chalcogramma*) and Japanese sandfish (*Arctoscopus japonicus*) were relatively low in our study, the two species were only distributed in the East Sea (Sea of Japan), constituting important fishery resources [44,45].

In our study, species richness and diversity tended to decrease with increasing water depth at both Ayajin and Hupo, but abundance did not show such a depth-related trend. Generally, the number of species, abundance (or biomass), and diversity of fish assemblages tend to decrease as water depth increases [46,47,48]. This decline is attributed to changes in environmental conditions such as temperature, pressure, light, and food availability, which can limit the types of species that can thrive at greater depths [49]. Furthermore, habitat complexity decreases with depth because of fewer complex structures, such as reefs or vegetated habitats. This reduction can lead to fewer niches available for different species, resulting in a lower diversity of fish assemblages [50]. In this study, the vertical profiles of water temperature showed distinct seasonal and spatial trends. In particular, the positions of the thermocline are located at the shallow layer in Ayjin and at the shallow and intermediate layers in Hupo. Fish assemblages in the thermocline zone may encounter wider ranges of water temperature as the seasons change, and consequently, this zone can be home to various types of fish, including those that prefer warmer or colder water [51]. Moreover, low values and narrow ranges of water temperature in the deeper layers may not be suitable for harboring a variety of fish species because fish species richness and diversity tend to increase in areas with a wider range of water temperatures than in regions with more homogeneous temperature conditions [52].

Multivariate analyses confirmed that the structure of demersal fish assemblages differed significantly between the two sites and among the three depth layers. Such differences were particularly evident between Ayajin and Hupo in the shallow layers and between the shallow and intermediate/deeper layers within each study site. The differences in assemblages may have been caused by variations in habitat use by individual fish species at different sites and depths. Canonical analysis of the principal coordinates revealed that two sculpin species (*A. elongatus* and *E. diceraus*) and one rockfish species (*Sebastes taczanowskii*) were limited to the shallow depth layer of Ayajin, whereas four demersal fish species (*C. pallasii*, *G. herzensteini*, *C. pinetorum*, and *Pseudopleuronectes herzensteini*) were highly associated with the shallow and intermediate depth layers of Hupo. In addition, *D. setiger*, *G. macrocephalus*, and *G. stelleri* contributed to fish assemblages in the intermediate and/or deeper layers of both Ayajin and Hupo. In the present study, the differences in demersal fish assemblages were likely associated with the habitat preferences of each species with respect to depth and water temperature. For example, among fish species, *C. pallasii* is distributed in the Northwest Pacific, but its main aggregations are formed at depths of less than 100 m [53], whereas Psychrolutidae fishes, including *Dasycottus setiger*, are typical deep-water fishes, and their vertical distributions are concentrated at depths greater than 150 m [4,54]. Three species, *A. elongatus*, *E. diceraus*, and *Sebastes taczanowskii*, are typical boreal species distributed at high latitudes in the northern hemisphere [55] and occur mainly at latitudes higher than 37° N, where the influence of the North Korea Cold Current is prevalent on the eastern coast of Korea [56].

In marine ecosystems, water temperature is widely regarded as the most significant environmental variable influencing species distribution and community structure within fish assemblages [52]. Based on the global water temperature patterns, the distributional ranges of fish species can be categorized primarily into tropical, subtropical, temperate, and boreal distributions, progressing latitudinally from low- to high-latitude regions [57]. The marine environment of the Korean Peninsula is classified within a typical temperate climate zone, characterized mainly by temperate species; however, subtropical species are also prevalent during the summer season and in sea areas at lower latitudes, such as Jeju Island [58,59]. According to marine zoogeography, the study area is situated in the western Pacific boreal region [60] and is home to several boreal marine fishes, such as Pacific cod and walleye pollock [61,62]. However, recent climate warming has led to shifts in the distribution of marine species, notably enabling poleward expansions in the distributional ranges of warm-water marine species and movement of the southern distributional boundary of cold-water species [63]. For example, in the Korean East Sea, the habitat distribution of walleye pollock has moved toward higher latitudes over the past few decades [64]. In addition, this study confirmed the occurrence of warm-water fish such as *Ditrema temminckii*, *Monocentris japonica*, *Konosirus punctatus*, and *Seriola quinqueradiata* [57], which were almost absent in a previous study conducted 15 years ago in the mid-latitudes of the Korean East Sea [17]. This variability in fish assemblages seems to correspond to recent warming trends in the East Sea [25]. Climate change trends in the East Sea are expected to become more evident and consequently lead to changes in the composition of fish communities; hence, continuous monitoring of fish communities is required to detect such variability.

## 5. Conclusions

In summary, this study offers insights into the spatial variability of demersal fish assemblages within the East Sea along the Korean coast. Our research highlights the significant impact of study site and depth on the assemblage structure of demersal fish, primarily influenced by variations in the abundance of common fish species. Moreover, we observed that species richness and diversity were significantly higher in the shallow layers than in the deeper layers. Although there were seasonal differences in the position of the thermocline, this study cannot detect any influences of seasonal thermocline depth on demersal fish assemblage. Nonetheless, since seasonal changes in thermocline can affect the vertical distribution of fish assemblage [65,66], it is important for further studies to dig deeper and confirm how much of an impact these seasonal shifts really have on fish assemblages. Understanding the composition of faunal assemblages in coastal habitats lays important baselines for future research and management strategies, particularly in regions like the eastern Korean waters, where research is currently limited. In addition, our findings hold promise for enhancing fisheries management in temperate coastal environments, which are home to diverse fishery resources and face challenges posed by climate change.

## Figures and Tables

**Figure 1 animals-14-01612-f001:**
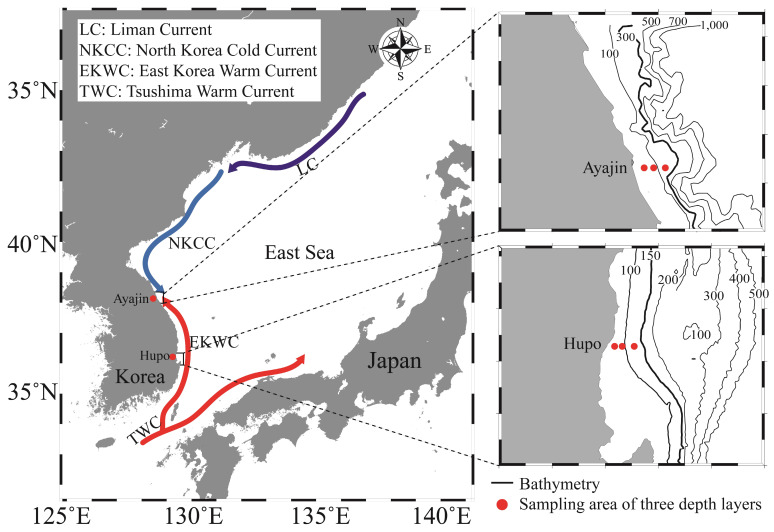
Map of the study area in the northern (Ayajin) and southern (Hupo) sites of the East Sea off the Korean coast. Fish samples were collected at three depth layers within the boxed areas.

**Figure 2 animals-14-01612-f002:**
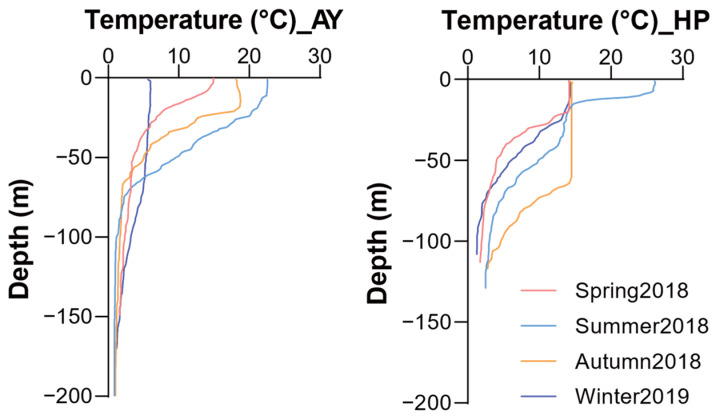
Vertical distribution of water temperature structures during four seasons in Ayajin (AY) and Hupo (HP).

**Figure 3 animals-14-01612-f003:**
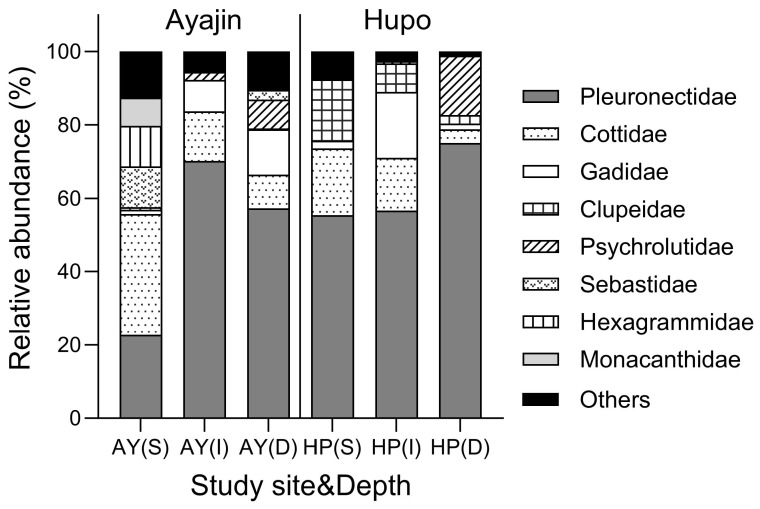
Relative abundance of fish species in each family with respect to study site and depth layer (AY, Ayajin; HP, Hupo; S, shallow; I, intermediate; D, deeper).

**Figure 4 animals-14-01612-f004:**
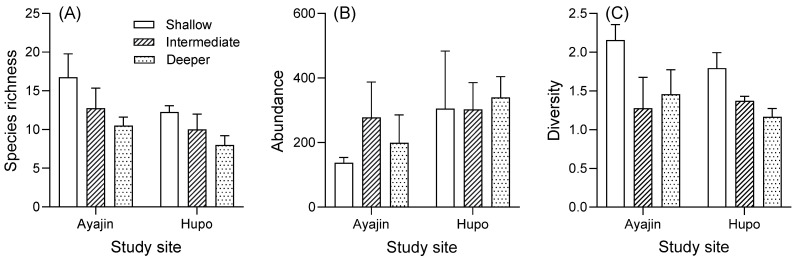
Variations in species richness (**A**), abundance (**B**), and diversity (**C**) of fish assemblage in relation to site and water depth.

**Figure 5 animals-14-01612-f005:**
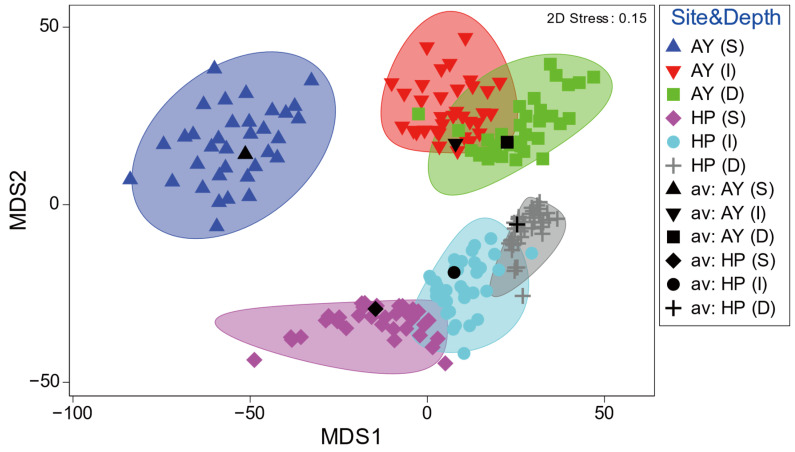
Spatial (site) and depth-related patterns of demersal fish assemblages by metric MDS ordination. Ellipses show approximate 95% confidence boundaries for average assemblages calculated by 999 bootstrap averaging across replicates with replacement (AY, Ayajin; HP, Hupo; S, shallow; I, intermediate; D, deeper).

**Figure 6 animals-14-01612-f006:**
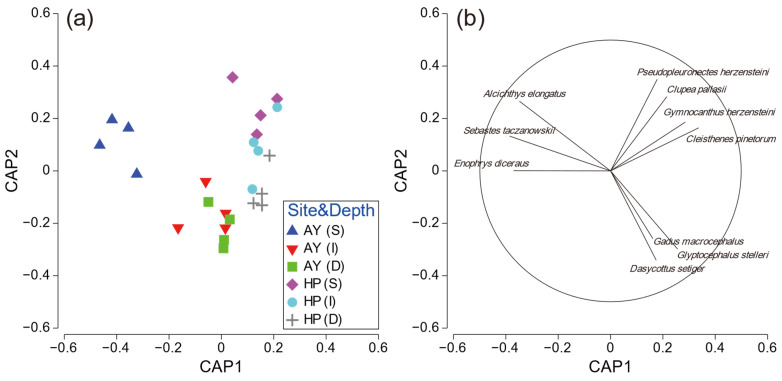
(**a**) Ordination plots for canonical analysis of principal coordinates of fish assemblage and (**b**) corresponding strength and direction of Pearson correlation in relation to site–depth interactions. AY, Ayajin; HP, Hupo; S, shallow; I, intermediate; D, deeper.

**Table 1 animals-14-01612-t001:** List of the most abundant fish families and species as a proportion of total abundance (%) collected from the Ayajin and Hupo coast of the East Sea, Korea.

Family	Ayajin (%)	Hupo (%)	Species	Ayajin (%)	Hupo (%)
Pleuronectidae	54.7	62.8	*Glyptocephalus stelleri*	50.0	23.9
Cottidae	16.3	11.8	*Cleisthenes pinetorum*	3.1	18.8
Gadidae	8.0	7.0	*Hippoglossoides dubius*	0.6	16.9
Clupeidae	0.2	8.7	*Gymnocanthus herzensteini*	2.6	11.2
Psychrolutidae	3.5	6.0	*Gadus macrocephalus*	7.6	7.0
Liparidae	1.5	2.2	*Clupea pallasii*	0.2	8.7
Sebastidae	4.4	<0.1	*Dasycottus setiger*	2.9	6.0
Hexagrammidae	2.5	0.3	*Icelus cataphractus*	5.0	0.2
Trichodontidae	2.7	0.2	*Pseudopleuronectes herzensteini*	0.4	2.7
Monacanthidae	1.7		*Alcichthys elongatus*	3.3	0.4
Cyclopteridae	1.6	<0.1	*Liparis ingens*	0.4	2.1
Other families	2.7	0.9	*Arctoscopus japonicus*	2.7	0.2
			*Hemilepidotus gilberti*	2.3	
			*Enophrys diceraus*	2.2	
			*Hexagrammos otakii*	1.6	0.1
			*Eumicrotremus orbis*	1.4	
			*Sebastes taczanowskii*	1.4	
			*Stephanolepis cirrhifer*	1.3	
			*Careproctus rastrinus*	1.1	<0.1
			*Sebastes owstoni*	1.1	
			Other species	8.8	2.0

**Table 2 animals-14-01612-t002:** Results of two-way ANOVA on the number of species, their abundance, and the diversity of fish assemblages in the study areas. Bold numbers indicate significant differences at *p* < 0.05.

Source	df	Species Richness		Abundance		Diversity	
		F	*p*	F	*p*	F	*p*
Site	1	12.733	**0.002**	5.319	**0.033**	2.663	0.120
Depth	2	11.412	**0.001**	1.276	0.303	14.703	**<0.001**
Site × Depth	2	0.058	0.943	0.529	0.598	1.554	0.239
Residual	24						

**Table 3 animals-14-01612-t003:** Results of pairwise PERMANOVA tests for site–depth interactions within each habitat, site, or season. Asterisk indicates significant difference at *p* < 0.05.

Site–Depth Interaction			
Site	Shallow	Intermediate	Deeper
Ayajin–Hupo	2.0252 *	1.3327	1.4139
Depth	Ayajin	Hupo	
Shallow–Intermediate	2.0040 *	1.1171	
Shallow–Deeper	2.3013 *	2.0174 *	
Intermediate–Deeper	0.8270	1.5104	

## Data Availability

Data are available on request due to restrictions.

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
