# Peer review of "Spatial Dynamics of Demersal Fish Assemblages off the Korean Coast in the East Sea"

_animals, 2024, doi:10.3390/ani14111612_

Round 1

Reviewer 1 Report

Comments and Suggestions for Authors

An interesting paper on distribution of fishes. I have some suggestions for improvement. 

Use of latitude to characterize the difference between two sites is problematic. Also, the difference is really based on temperature and the interacting currents rather than the gradual effect of latitude on temperature. Please characterize these two sites in a different way than by latitude.

Don’t use acronyms for naming water currents. Just spell it out every time. Acronyms are a sloppy way of making sure your paper is not understandable.

Please make an inset of the entire west Asian area for the map, so those unfamiliar with Korea can understand where your study is located. Please include the water currents and their path on the map as well.

Line 115 what is CTD? Spell it out and explain on first use.

Line 122 significant differences of what? What is your response variable and how are you comparing the response?

Line 130 what do you mean by all species. Are you saying every site was assumed to have the same species and thus same species richness?

Line 131 what does x signify here?

Line 137 I think this should be “non-metric” multidimensional scaling, nMDS

Overall, the methods are incomplete and not clear. Try starting by explaining what about the distribution of the fishes you are testing. Then tell how your chosen statistical test helps to understand your biological question. Explain the test and how it works. For example, what is PERMANOVA and Canonical Analysis of Principle Coordinates? What results from these tests would suggest similarity or significance?

Why such effort spent on characterizing seasonal changes in water temperature when the seasonal factor was not significant for the fish assemblage data? Are you describing differences in fish assemblages or temperatures?

Comments on the Quality of English Language

I applaud the authors for their obvious work on English language quality. However, there are still several places where the meaning is unclear because of improper usage. I suggest you have someone read through and help with English editing.

Author Response

Reviewer #1

Comments and Suggestions for Authors

An interesting paper on distribution of fishes. I have some suggestions for improvement.

Use of latitude to characterize the difference between two sites is problematic. Also, the difference is really based on temperature and the interacting currents rather than the gradual effect of latitude on temperature. Please characterize these two sites in a different way than by latitude.

(Response) We changed the use of “latitude” into “oceanic current region”, because two study sites are located on the different path of oceanic currents, i.e., the Hupo area is located on the path of the East Korea Warm Current, whereas the Ayajin is located on the boundary between the East Korea Warm Current and the North Korean Korea Cold Current (see also Figure 1). We added a description of why we selected two different study sites in the “2.1 Study Area” section.

Don’t use acronyms for naming water currents. Just spell it out every time. Acronyms are a sloppy way of making sure your paper is not understandable.

(Response) We used full naming water currents including acronyms when the names were the first referred in the text, and the spell them out with acronyms thereafter.

Please make an inset of the entire west Asian area for the map, so those unfamiliar with Korea can understand where your study is located. Please include the water currents and their path on the map as well.

(Response) We revised the study map by including the entire west Asian area and the paths of main currents around Korean Peninsular.

Line 115 what is CTD? Spell it out and explain on first use.

(Response) CTD indicated conductivity, temperature and depth. So we moved CTD in the infront of this sentence.

Line 122 significant differences of what? What is your response variable and how are you comparing the response?

(Response) We revised this sentence by specifying response variables for identifying significant differences.

Line 130 what do you mean by all species. Are you saying every site was assumed to have the same species and thus same species richness?

(Response) “All species” means that the ANOVA test used all species collected in this study. Because we thought this phrase is unnecessary, we removed it from the sentence.

Line 131 what does x signify here?

(Response) The “x” indicates the abundance of each fish species. So, we changed log(x+1) into log(abundance+1) accordingly.

Line 137 I think this should be “non-metric” multidimensional scaling, nMDS

(Response) We would retain the figure of mMDS in the revision, because the result can identify how much multivariate dispersions were overlapped with the depths within each site.

Overall, the methods are incomplete and not clear. Try starting by explaining what about the distribution of the fishes you are testing. Then tell how your chosen statistical test helps to understand your biological question. Explain the test and how it works. For example, what is PERMANOVA and Canonical Analysis of Principle Coordinates? What results from these tests would suggest similarity or significance?

(Response) We added a description of the distribution of the fish we are testing. We selected two study sites to identify how the distribution of fish assemblages varied depending on the path of sub-currents (the East Korea Warm Current and the North Korea Cold Current) and water depth in the East Sea. We also added some descriptions of why we used several multivariate methods to analyze dynamics of fish assemblage structures. The changes were made in the sections of “2.1 Study Area” and “2.3 Data Analyses”.

Why such effort spent on characterizing seasonal changes in water temperature when the seasonal factor was not significant for the fish assemblage data? Are you describing differences in fish assemblages or temperatures?

(Response) Although we conducted seasonal samplings, we did not find any significant seasonal differences in fish assemblage structures. Seasonal changes in water temperature were only evident in shallower depth layers, but the water temperature was almost constant in the deeper layers. This wider range of water temperature throughout the year may allow diverse fishes occurring in shallow layers. We added some descriptions about water temperature in the second paragraph in Discussion section.

Comments on the Quality of English Language

I applaud the authors for their obvious work on English language quality. However, there are still several places where the meaning is unclear because of improper usage. I suggest you have someone read through and help with English editing.

(Response) Thank you for the good evaluation on English quality. Although the manuscript has was English editing from the language consultant company (i.e., https://www.editage.com/), we noticed that there also were some poorer languages in the text. Because the deadline of resubmission is not to long, we will get another English editing during next round revision.

Reviewer 2 Report

Comments and Suggestions for Authors

The manuscript entitled “Spatial Dynamics of Demersal Fish Assemblages off the Korean Coast in the East Sea” deals with the topic of latitudinal variation of fish assemblage completing with variation in depth at same latitude. Latitudinal variation is a popular hypothesis that has been tested in many species with topics such as reproduction timing, growth areas, nursing areas or as in the case of this manuscript, fish assemblage. But this hypothesis remains poorly understood in many cases as the present study is trying to address. Some studies suggest indirectly that temperature is an important environmental stimulus (proximal factor) for some species. The hypothesis was well supported, because a condition of relatively constant temperatures in tropical seas, with high water temperatures year-round, is coupled with some species aggregation. The authors look forward to explaining what is happening in Korean seas with fish assemblage at two latitudes and three different depths. This is the relevant part of the study because it contributes to filling the gap of fish assemblage knowledge in this area.

The study considered the hypothesis of global warming that is affecting the sea temperature starting with the idea that in marine ecosystems, water temperature is widely regarded as the most significant environmental variable influencing species distribution and community structure within fish assemblages. The hypothesis is linked with the other idea that global warming is triggering shifts in the distribution of marine species. Particularly is assumed a northward migration of tropical fishes. The study is telling us that in the Korean Peninsula, that is classified within a typical temperate climate zone, should be composed of temperate species; however, subtropical species are also prevalent during the summer season. For this reason, the contributions become useful for the academic community.

In conclusion, the topic of the manuscript is important for the scientific community and the authors made a good design in order to accomplish the fish variability along latitudinal gradient and depth gradient. In my opinion the contribution is well enhanced.

I have some minor concerns.

In figure 2 I suggest a better contrast of the colors. Actual colors are difficult to follow. Summer 2018 and winter 2019 resulted difficult to me.

In line 179 a “D” is missed in word Demersal

Author Response

Reviewer #2

Comments and Suggestions for Authors

The manuscript entitled “Spatial Dynamics of Demersal Fish Assemblages off the Korean Coast in the East Sea” deals with the topic of latitudinal variation of fish assemblage completing with variation in depth at same latitude. Latitudinal variation is a popular hypothesis that has been tested in many species with topics such as reproduction timing, growth areas, nursing areas or as in the case of this manuscript, fish assemblage. But this hypothesis remains poorly understood in many cases as the present study is trying to address. Some studies suggest indirectly that temperature is an important environmental stimulus (proximal factor) for some species. The hypothesis was well supported, because a condition of relatively constant temperatures in tropical seas, with high water temperatures year-round, is coupled with some species aggregation. The authors look forward to explaining what is happening in Korean seas with fish assemblage at two latitudes and three different depths. This is the relevant part of the study because it contributes to filling the gap of fish assemblage knowledge in this area.

The study considered the hypothesis of global warming that is affecting the sea temperature starting with the idea that in marine ecosystems, water temperature is widely regarded as the most significant environmental variable influencing species distribution and community structure within fish assemblages. The hypothesis is linked with the other idea that global warming is triggering shifts in the distribution of marine species. Particularly is assumed a northward migration of tropical fishes. The study is telling us that in the Korean Peninsula, that is classified within a typical temperate climate zone, should be composed of temperate species; however, subtropical species are also prevalent during the summer season. For this reason, the contributions become useful for the academic community.

In conclusion, the topic of the manuscript is important for the scientific community and the authors made a good design in order to accomplish the fish variability along latitudinal gradient and depth gradient. In my opinion the contribution is well enhanced.

(Response) We appreciate the reviewer’s good evaluation of our paper. In this paper, we analyzed spatio-temporal dynamics of demersal fish assemblage through fish samplings from two sites across three depth layers and during four seasons. Our study was the first to attempt to see how fish assemblage on the Korean Peninsula changes with depth, but there was no seasonal difference. As indicated by reviewer, we hope to our paper making useful contribution for the academic community.

I have some minor concerns.

In figure 2 I suggest a better contrast of the colors. Actual colors are difficult to follow. Summer 2018 and winter 2019 resulted difficult to me.

(Response) We revised color patterns of seasonal depth profile of temperature as suggested.

In line 179 a “D” is missed in word Demersal

(Response) We revised this word as suggested.

Reviewer 3 Report

Comments and Suggestions for Authors

The work is interesting but should be improved before publication.

1. A simple Summary is too simple. The authors use “different types of fish” instead of “fish species”, which is confusing (I don't think anyone doesn't know what a fish species is). I also suggest to mention more fish species here (using their common names).

2. Abstract. With the fish Glyptocephalus stelleri It is unclear: “In Ayajin, Glyptocephalus stelleri … predominated, whereas G. stelleri …were prevalent in Hupo”(?) “In Ayajin, G. stelleri dominated in both intermediate and deeper layers, …. Conversely, in Hupo, G. stelleri, dominated across all depth layers” (Isn’t this almost the same things?).

3. In the Introduction, several water currents are noted, but the Liman Cold Current (LCC) is mentioned only in the Materials and Methods. Say a few words about LCC in the Introduction.

 It would be useful to show the main currents in Figure 1.

4. Study Area. The difference between the two sites is that the width of the continental shelf is quite different: in the Ayajin area it is much narrower than in the southern Hupo area, and the 1000-m slope isobath in Ayajin is located at a distance from the coastline where there is a 300meter plateau in Hupo area. This should affect fish communities much, which is not noted and discussed in your paper.

5. “Field sampling was conducted seasonally from May 2018 to March 2019 “ – but more precisely, Field sampling was conducted over a year, seasonally from May 2018 to March 2019.

6. Results. The title “Changes in Vertical Trends of Water Temperature between 2018 and 2019 does not reflect the essence of the study. You are not comparing two years (having 3 seasons in 2018 and one season in 2019), you are comparing Water Temperature trends of annual circle (spring/summer/autumn/ winter) in two localities, northern and southern.

7. In Figure 1, for each region A and H, three red dots are shown where oceanographic monitoring was carried out, and at different depths. Figure 2 does not explain which points the graphs are for. This is also not explained in the Methods section. Please clarify their position.

8. When studying water masses and fish assemblages using water profiles such as yours, they usually discuss the position of the thermocline that is a transition layer (an abrupt change in temperature) between the warmer mixed water at the sea surface and the cooler deep water below; its depth varies depending on the season.

As I can see, in Ayajin the thermocline is at a depth about 40 m in spring; and ca. 60-70 m in summer and autumn (in winther the water is almost homogenous in T). Below 60-70 m throughout the year, the water temperature is consistently below 5 C. In Hupo the temperature dynamic is clearly different. Discuss whether the differences depend on the width of the shallow shelf at the two sites or on currents.

9. Nothing is said about the water salinity. Nothing is said about grounds or soils (this is important as Pleuronectids predominated).

10. Figure 4. Designate A, B and C in this picture.

11. Fish Species Composition. You listed 20 species out of 73. Where are the rest? The value of the article (and its citation) will increase if you provide complete listings in two places

12. Discussion.  In our study, species richness, abundance, and diversity varied with the study site and water depth”. This is a banal conclusion. It might be more interesting if you can discuss also relations with other characteristics of two localities (shelf width, differences in warm upper water layer in autumn, currents, salinity, grounds etc.).

Dissolved organic carbon (DOC) is unlikely to be discussed as it is not addressed in this study.

13. “The study area is situated at the northern end of the temperate oceanic climate zone and is home to several cold-water marine fishes, such as Pacific cod and walleye pollock, originating from the boreal oceans [57,58]”.

Marked in bold is not true. You should cite zoogeographic publications (See at least Briggs, J.C. Marine Zoogeography). The walleye pollock and Pacific cod are typical boreal species.

General conclusion: the MS needs improvement. Reconsider after major revision

Author Response

Reviewer #3

Comments and Suggestions for Authors

The work is interesting but should be improved before publication.

  1. A simple Summary is too simple. The authors use “different types of fish” instead of “fish species”, which is confusing (I don't think anyone doesn't know what a fish species is). I also suggest to mention more fish species here (using their common names).

(Response) In the Simple Summary section, we changed “different types of fish” into “fish species” to avoid any confusion between the two words. We also indicated the most abundant three fish species in each latitude including their common names, but we did not mentioned more fish species due to word limit in this section (According to instructions for authors, the journal recommend the the simple summary consists of no more than 200 words)

  1. Abstract. With the fish Glyptocephalus stelleri It is unclear: “In Ayajin, Glyptocephalus stelleri … predominated, whereas G. stelleri …were prevalent in Hupo”(?) “In Ayajin, G. stelleri dominated in both intermediate and deeper layers, …. Conversely, in Hupo, G. stelleri, dominated across all depth layers” (Isn’t this almost the same things?).

(Response) To clarify the sentences that the reviewer is confused, we revised the first sentence into “Although Glyptocephalus stelleri was the most abundant fish species in both Ayajin and Hupo, Gadus macrocephalus, Icelus cataphractus, and Alcichthys elongatus were mort predominat in Ayajin, whereas Cleisthenes pinetorum, Hippoglossoides dubius, and Gymnocanthus herzen-steini were more prevalent in Hupo.” We also added “In terms of depth layer,” in the beginning of the second sentence.

  1. In the Introduction, several water currents are noted, but the Liman Cold Current (LCC) is mentioned only in the Materials and Methods. Say a few words about LCC in the Introduction.

(Response) We mentioned Liman Cold Current (LCC) in the second paragraph of the Introduction as suggested.

It would be useful to show the main currents in Figure 1.

(Response) We added the paths of main currents in Figure 1.

  1. Study Area. The difference between the two sites is that the width of the continental shelf is quite different: in the Ayajin area it is much narrower than in the southern Hupo area, and the 1000-m slope isobath in Ayajin is located at a distance from the coastline where there is a 300meter plateau in Hupo area. This should affect fish communities much, which is not noted and discussed in your paper.

(Response) Unfortunately, it is hard to discuss the influences of the location and width of the continental shelf on fish assemblage. In addition, since we did not observe the fish assemblage at depth deeper than 150 m, we cannot discuss about this issue. As suggested by the reviewer, we will conduct further studies on fish assemblage in relation to bottom topography through samplings at wider depth ranges.

  1. “Field sampling was conducted seasonally from May 2018 to March 2019 “ – but more precisely, Field sampling was conducted over a year, seasonally from May 2018 to March 2019.

(Response) We revised this sentence as suggested.

  1. Results. The title “Changes in Vertical Trends of Water Temperature between 2018 and 2019” does not reflect the essence of the study. You are not comparing two years (having 3 seasons in 2018 and one season in 2019), you are comparing Water Temperature trends of annual circle (spring/summer/autumn/ winter) in two localities, northern and southern.

(Response) We changed the subtitle of 3.1 into “Vertical Trends of Water Temperature in northern and southern sites” according to reviewer’s comment.

  1. In Figure 1, for each region A and H, three red dots are shown where oceanographic monitoring was carried out, and at different depths. Figure 2 does not explain which points the graphs are for. This is also not explained in the Methods section. Please clarify their position.

(Response) Because the distance between three sampling locations covering three depth layers is considerably close, we conducted CTD observations only at the deepest location. We also added this explanation in the Methods section.

  1. When studying water masses and fish assemblages using water profiles such as yours, they usually discuss the position of the thermocline that is a transition layer (an abrupt change in temperature) between the warmer mixed water at the sea surface and the cooler deep water below; its depth varies depending on the season.

As I can see, in Ayajin the thermocline is at a depth about 40 m in spring; and ca. 60-70 m in summer and autumn (in winther the water is almost homogenous in T). Below 60-70 m throughout the year, the water temperature is consistently below 5 C. In Hupo the temperature dynamic is clearly different. Discuss whether the differences depend on the width of the shallow shelf at the two sites or on currents.

(Response) We added some descriptions about the position of the thermocline in the second paragraph of Discussion section. However, we did not discuss the influences of seasonal thermocline on fish assemblage because there are no seasonal differences in fish assemblages in both study sites. Some previous studies noted that seasonal thermocline affects the vertical distribution of fish assemblage, so we suggested this issue for future studies in the Conclusion section. In addition, since we did not analyze the influences of the width of the shelf and oceanic current on fish assemblage, we did not mention this issue in the manuscript. We hope to validate this issue in future studies.

  1. Nothing is said about the water salinity. Nothing is said about grounds or soils (this is important as Pleuronectids predominated).

(Response) In this study, the salinities are almost constant seasonally in the sampling depth (below 50 m). Unfortunately, we did not analyze sediment composition in the study area and there also was a lack of available literature for sediment characteristics around study sits. Thus, we did not mention the salinity and sediment in this manuscript.

  1. Figure 4. Designate A, B and C in this picture.

(Response) We designated A, B, and C in Figure 4 as well as its caption.

  1. Fish Species Composition. You listed 20 species out of 73. Where are the rest? The value of the article (and its citation) will increase if you provide complete listings in two places.

(Response) We want to show only the top dominant species among all fish species collected, so we listed only 20 species constituting >1% of total abundance. Instead, we attached the list of all species as supplementary material (i.e., Table S1).

  1. Discussion. “In our study, species richness, abundance, and diversity varied with the study site and water depth”. This is a banal conclusion. It might be more interesting if you can discuss also relations with other characteristics of two localities (shelf width, differences in warm upper water layer in autumn, currents, salinity, grounds etc.).

(Response) We revised this paragraph according to reviewer’s suggestions. In particular, we added some discussion about range of water temperature and fish diversity in each depth layer. However, we cannot discuss fish diversity in relation to shelf width, current, salinity and sediment due to the lack of available information about these characteristics in the study area. We want to remain these issues for future studies.

Dissolved organic carbon (DOC) is unlikely to be discussed as it is not addressed in this study.

(Response) We removed the descriptions about DOC from the Discussion section.

  1. “The study area is situated at the northern end of the temperate oceanic climate zone and is home to several cold-water marine fishes, such as Pacific cod and walleye pollock, originating from the boreal oceans [57,58]”.

Marked in bold is not true. You should cite zoogeographic publications (See at least Briggs, J.C. Marine Zoogeography). The walleye pollock and Pacific cod are typical boreal species.

(Response) We revised the sentence as suggested. According to “Marine Zoogeography”, our study area is included to the “Western Pacific Boreal Region”, but we found that the study area gradually changed into the temperate ocean due to recent climate change. Consequently, we revised the sentence as follows.

“The study area is situated at the western Pacific boreal region [56] and is home to several boreal marine fishes, such as Pacific cod and walleye pollock [57,58].”

General conclusion: the MS needs improvement. Reconsider after major revision

(Response) We tried to revise the paper by reflecting the reviewer’s comments as much as possible. We hope that our revised manuscript will improve much more than the previous version.

Round 2

Reviewer 3 Report

Comments and Suggestions for Authors

The manuscript can be accepted in present form